# TiNi Alloy Lattice Structures with Negative Poisson's Ratio: Computer Simulation and Experimental Results

**Eduard Farber \*, Alexey Orlov, Evgenii Borisov, Arseniy Repnin, Stepan Kuzin, Nikita Golubkov and Anatoly Popovich**

Institute of Machinery, Materials, and Transport, Peter the Great St. Petersburg Polytechnic University (SPbPU), Polytechnicheskaya, 29, 195251 Saint Petersburg, Russia
* Correspondence: d.farber2010@yandex.ru

**Abstract:** One of the issues that modern implants face is their high stiffness, coupled with a positive Poisson's ratio along the implant. This creates certain problems with bone inflammation and implant detachment. A possible solution to these problems is TiNi alloy lattice structure implants with low stiffness and negative Poisson's ratio. This paper presents the results of simulation, fabrication by the SLM technique, and study of lattice structures with negative Poisson's ratio, which can help to solve said problems. The studies involve the determination of mechanical characteristics, Poisson's ratio, transformation temperatures, and the potential for a superelasticity effect of the lattice structure. The characteristics obtained at initial simulation were partially confirmed in the course of the works. Moreover, the possibility of fabricating TiNi alloy lattice structures with negative Poisson's ratio (about $-0.00323$) and low Young's modulus values (0.818 GPa) was confirmed by the SLM technique.

**Keywords:** lattice structure; negative poisson's ratio; nitinol; shape memory alloy; superelasticity





## 1. Introduction

Nitinol, a binary alloy of titanium and nickel, is one of the best-known smart materials. This is a temperature-sensitive alloy capable of realizing reversible diffusion-free thermoelastic phase transformations between two phases: high-temperature austenitic and low-temperature martensitic. The alloy can realize shape memory effect and superelasticity effect due to the thermoelastic transformation phenomenon [1,2]. This alloy was patented in 1965 and used in the industry for the first time in 1969 as a material for thermomechanical coupling of pipelines in hydraulic systems of F-14 fighters [3,4].

Further development of science and technology has expanded the Nitinol application. The alloy is currently applied in automotive, aerospace, and other industries as a material for thermal actuators and other devices of various purposes [3,5–8]. Nitinol has also found its application in various fields of medicine due to its excellent biocompatibility because of the presence of oxide layers on the surface, consisting mainly of titanium dioxide [9–13]. In orthopedics, Nitinol is used as a material for medical staples and other implants to treat various diseases of the skeletal system. In dentistry, the alloy is used as braces, in cardiology—as stents, artificial heart valves, pacemakers, and many other medical products [3,14–17].

The development of additive technologies allows expanding Nitinol application as a smart material. Additive technologies make it possible to fabricate products with complex geometry that cannot be produced by classical machining methods. This may help to solve a number of problems including those in the medical sphere. In addition, it is noted that the use of additive technologies does not have a negative impact on the level of biocompatibility of the alloy. It is noted that the use of additive technologies with laser scanning (for example, SLM) improves the level of biocompatibility of the resulting structures, due to a thicker oxide film on the surface [18,19].

One of these problems is related to implants for various purposes. Today, most implants are fabricated from materials with stiffness higher than that of human bone tissue.

The elastic modulus of a human cortical bone is 12–18 GPa, while the elastic modulus of a cancellous (trabecular) bone equals 0.1–5 GPa [20–25]. In turn, the elastic modulus of titanium and its alloys, of which most implants are made, is several times higher: 102–104 GPa for pure titanium, 110–114 GPa for Ti-6Al-4V alloy, and 50 to 100 GPa for other applied titanium alloys [20,25–28].

Due to high stiffness, an implant carries a significant body load, which reduces a load on the bone around the implant. Reduced bone loading, in turn, can lead to bone resorption at the bone–implant interface, ingress of wear particles with subsequent inflammation and gradual loosening and detachment of the implant. In the literature, this phenomenon is known as stress shielding effect [22,29,30]. There is a possibility of circulation of wear particles in the body (with their detection in the bone marrow, lymphatic system), followed by the induction of cytotoxicity and neoplasms [28,31].

This problem can be solved by reducing stiffness of the implant material to a level comparable to the stiffness of a human bone. Nitinol is one of the promising materials in this field. The Young's modulus of Nitinol in martensite is 30–40 MPa, in austenite—75–83 MPa [3,14]. These values are clearly lower than those of the Ti-6Al-4V alloy, but still exceed the elastic modulus of a human bone. As an alternative, the Young's modulus of Nitinol can be reduced using lattice structures with a certain level of porosity, which are fabricated by the SLM technique. An increased porosity significantly reduces the Young's modulus of the fabricated structures. In [22], this idea was confirmed by computer simulation. This approach has been described in papers [30,32–34] where lattice structures with low elastic modulus values were obtained through the use of porous structures with the geometry of single cells based on struts with different levels of porosity. Values of elasticity modulus have been obtained close to those of the human cortical bone. However, values were not achieved close to those of the human trabecular bone.

The existence of a relationship between the topology of a unit cell of the lattice structure and the properties of the alloy was noted in [30]. Furthermore, in [24], it was noted that a computer simulation can be implemented to predict the behavior of TiNi samples with different levels of porosity and geometry. In turn, in our work [35], using computer simulation, the theoretical possibility of controlling the properties of the lattice structure, including the values of the modulus of elasticity, was confirmed not only by changing the porosity, but also by changing the geometry of a unit cell of the lattice structure.

Apart from the implant stiffness problem, there is also the problem of implant detachment due to the constant positive Poisson's ratio along the implant. When the implant with a constant positive Poisson's ratio is used under cyclic compression–extension loading conditions, one side of the implant will be pressed against the bone at the interface, while the other, on the contrary, will be drawn aside. The latter side is more susceptible to surface damage. Besides, the implant drawn aside allows wear particles to penetrate into the space between the implant and the bone, causing the patient's immune system to react to the foreign body, which results in inflammatory bone loss [29,36].

This problem can be solved using complex implants, where the side drawn away from the bone will have a negative Poisson's ratio. Due to this, the side will exert additional pressure on the bone-implant interface when used cyclically, mechanically stimulating bone growth and blocking the possibility of bone tissue detachment and inflammation [36].

The priority task can be clearly defined based on the above-described problems: development and fabrication of TiNi alloy lattice structures with low elastic modulus and negative Poisson's ratio. The solution to this problem must be considered in a complex way, using the possibilities of computer simulation for the initial programming of the properties of the lattice structure, and production of this lattice structure by the SLM method from TiNi alloy powder. Accordingly, it is necessary to develop a complete methodology for obtaining lattice structures from the TiNi alloy, which makes it possible to obtain products for various purposes with different properties.

This paper presents an integrated approach to solving the described problem, consisting of the implementation of several successive stages: computer simulation of a lattice structure

for programming its characteristics; fabrication of a lattice structure by the SLM method from TiNi alloy powder; study of the resulting lattice structures. The study involves the determination of mechanical characteristics and the possibility to implement the superelasticity phenomenon useful in the advanced use of the fabricated structures in implants subjected to cyclic loads. Besides, the study allows checking the simulated parameters of lattice structures for compliance with the actual parameters and the general capability to fabricate TiNi alloy lattice structures with negative Poisson's ratio by the SLM technique. Thus, the take home message of this work is the experimental confirmation of the developed technique for obtaining lattice structures from the TiNi alloy with programmable properties—a low value of the elastic modulus and a negative Poisson's ratio.

## 2. Materials and Methods

Figure 1 shows the block diagram that explain the workflow of the present study, consisting of several main steps—computer simulation, fabrication, and study of the resulting lattice structure.

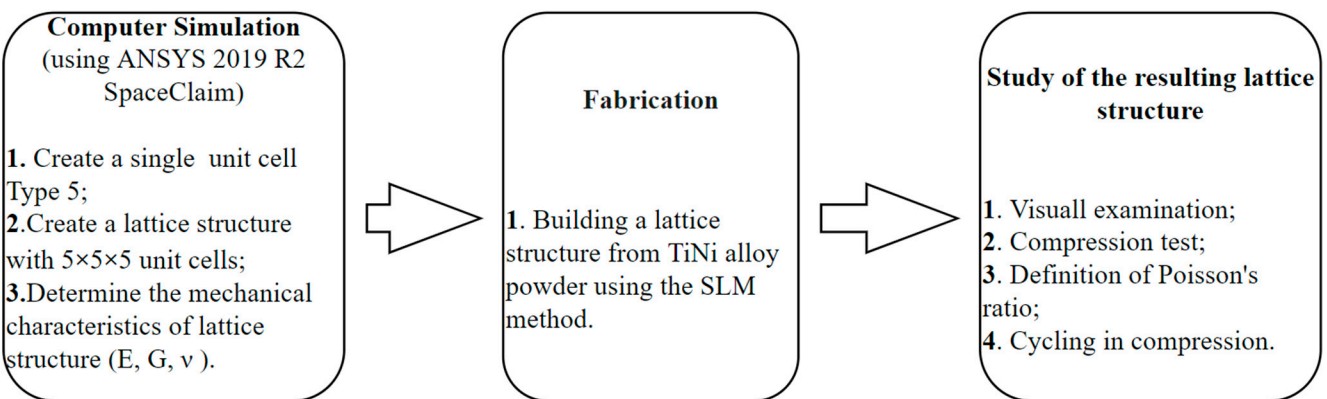

**Figure 1.** Block diagram of the study.

Samples were fabricated from spherical powder of TiNi alloy (with atomic percent of 49% and 51% for Ti and Ni, respectively) by CNPC Powder, Shanghai, China, using the electrode induction melting gas atomization method (EIGA). A SEM image of the alloy powder is shown in Figure 2a. Figure 2b presents the powder particle size.

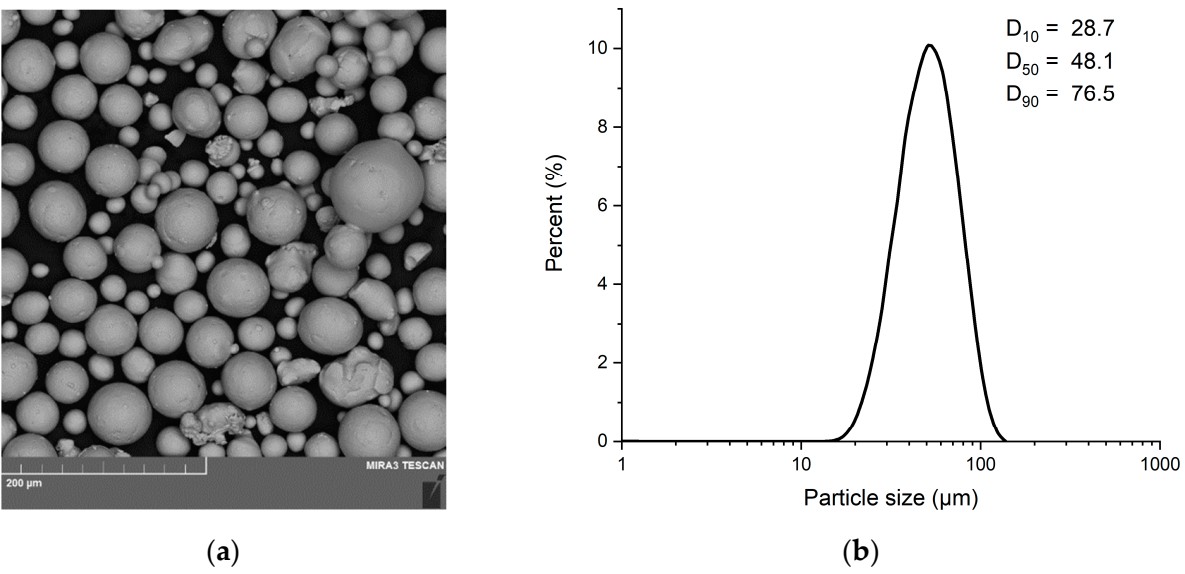

**Figure 2.** (**a**) SEM image of initial powder; (**b**) Initial powder particle size.

A single unit cell with negative Poisson's ratio was simulated with the ANSYS 2019 R2 SpaceClaim software package (ANSYS 2019 R2, Ansys Inc., Canonsburg, PA, USA). This single unit cell topology was named Type 5. The single unit cell has dimensions of 2 mm × 2 mm × 2 mm and 80% porosity. A lattice structure consisting of 5 unit cells on three axes, i.e., 5 × 5 × 5 cells with total dimensions of 10 mm × 10 mm × 10 mm, was generated on the basis of the simulated single unit cell. A Type 5 single cell with negative Poisson's ratio and the lattice structure itself are shown in Figure 3a,b, respectively. The mechanical characteristics of the simulated lattice structure with 5 × 5 × 5 unit cells (Young's modulus (E), shear modulus (G), and Poisson's ratio) were determined by computer simulation of compression experiment with static load in elastic region, using ANSYS 2019 R2 SpaceClaim software package. Initial simulation data: Young's modulus of austenite—77 GPa, Young's modulus of martensite—30 GPa, Poisson's ratio—0.33, shear modulus—29 GPa, density—6500 kg/m$^3$. To determine Young's modulus, the normal displacement boundary condition without friction was applied to the lower faces of the lattice structure and the small displacement boundary condition was applied to the upper faces. For shear modulus, the normal displacement boundary condition without friction was applied to the lower and upper faces of the lattice structure, and the small displacement boundary condition was applied to the side faces.

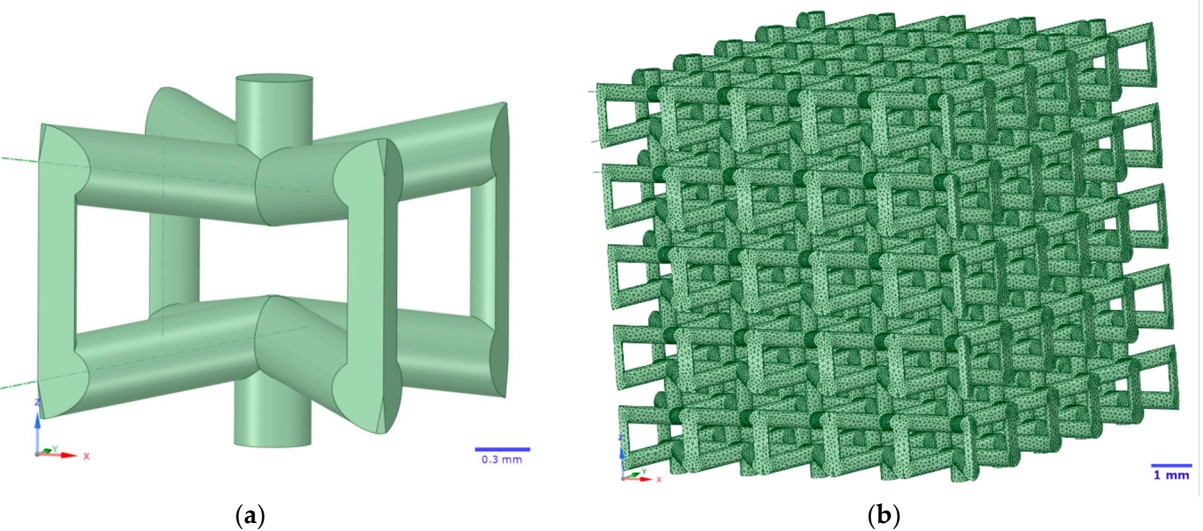

(**a**)                              (**b**)

**Figure 3.** (**a**) Single Type5 unit cell with negative Poisson's ratio; (**b**) Lattice structure based on a single Type 5 unit cell.

The structures were processed by the Materialise Magics software package (Magics 25, Materialise NV, Leuven, Belgium) for fabrication and preparation for mechanical and other tests. The bases sized 10 mm × 10 mm × 6 mm and 10 mm × 10 mm × 3 mm were added to the lattice structure at the top and bottom.

Six lattice structures (3 structures with 10 mm × 10 mm × 6 mm bases and 3 structures with 10 mm × 10 mm × 3 mm bases) were fabricated by the SLM technique using the SLM280HL system (SLM Solutions GmbH, Lübeck, Germany). The system is equipped with an ytterbium fiber laser with a maximum power of 400 W, a wavelength of 1070 nm, a minimum laser beam diameter of 80 μm, and a maximum scanning rate of 15 m/s. The structures were fabricated in inert gas atmosphere (argon). The fabrication process parameters: laser power—200 W, scanning rate—925 mm/s, distance between laser passes—0.08 mm, layer thickness—0.03 mm.

Porosity of the fabricated lattice structures was calculated by formula (1) [25,37,38]:

$$P = (1 - p^*/p_s), \tag{1}$$

where, p* is lattice structure density, $p_s$ is density of the material from which the lattice structure was fabricated.

The initial powder and the fabricated lattice structures were visually examined using TESCAN Mira 3 LMU (TESCAN, Brno, Czech Republic) scanning electron microscope (SEM) with secondary electrons (SE) and backscattered electrons (BSE). Phase composition of the initial powder and the fabricated samples was determined using Bruker D8 Advance X-ray diffraction (XRD) meter (Bruker, Bremen, Germany), using Cu-K$\alpha$ ($\lambda$ = 1.5418 Å). Transformation temperatures were determined using a high-temperature differential scanning calorimeter DSC 404 F3 Pegasus by NETZSCH-Gerätebau GmbH, Selb, Germany. The heating rate in argon atmosphere was 5 K/min.

Compression test and cycling in compression were carried out using a Zwick/Roell Z100 single-axis floor-mounted testing machine (Zwick/Roell, Ulm, Germany). The compression direction was perpendicular to the building direction (BD). Cycling loading stress varies from 9 to 20 Mpa. An example of reversible and irreversible strain during cycling under the superelasticity effect is shown in Figure 4. A Gleeble 3800 testing machine (Dynamic Systems Inc., Austin, TX, USA) with a strain gauge type hot zone transducer (Hot Zone L-Strain) and crosswise LVDT type gauge (C-Gauge) was used to determine longitudinal strain and lateral strain of compressed samples for Poisson's ratio calculation. The Poisson's ratio was calculated by formula (2) [39] based on the obtained data:

$$\nu = -\varepsilon_t / \varepsilon \tag{2}$$

where, $\varepsilon$ is relative longitudinal (axial) strain of the sample calculated as $\delta l/l$ (l is sample reference length), $\varepsilon_t$ is relative lateral strain of the sample calculated as $\delta b/b$ (b is sample width) [40].

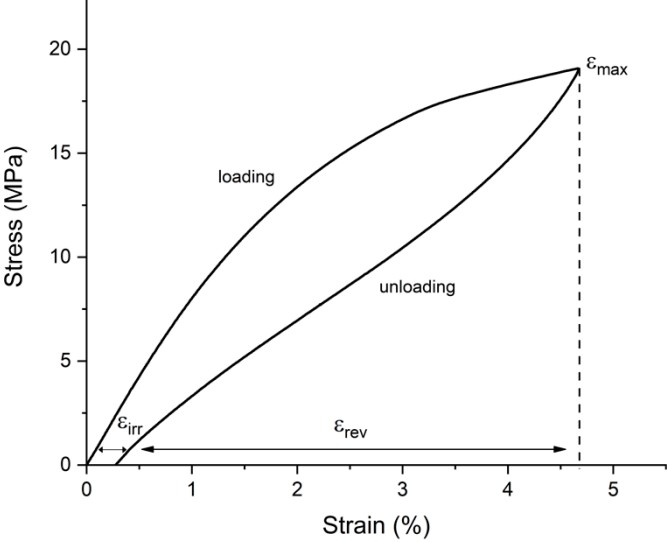

**Figure 4.** Maximum, reversible, and irreversible (residual) strain determined under the superelasticity effect.

## 3. Results and Discussion

### 3.1. Results of Lattice Structures Simulation and Growth

The mechanical characteristics given below were determined for the lattice structure with the single unit cell Type 5 topology during initial simulation of the compression experiment: Young's modulus (E)—0.61 GPa, shear modulus (G)—3.68 GPa, Poisson's ratio ($\nu$)——0.13. Diameter of the lattice structure struts at its 80% porosity was 420 μm.

Then, lattice structures were fabricated by the SLM technique. Samples were built in the horizontal position. Figure 5a shows orientation of a sample and its building direction (BD). Figure 5b shows a fabricated lattice structure sample in the vertical position.

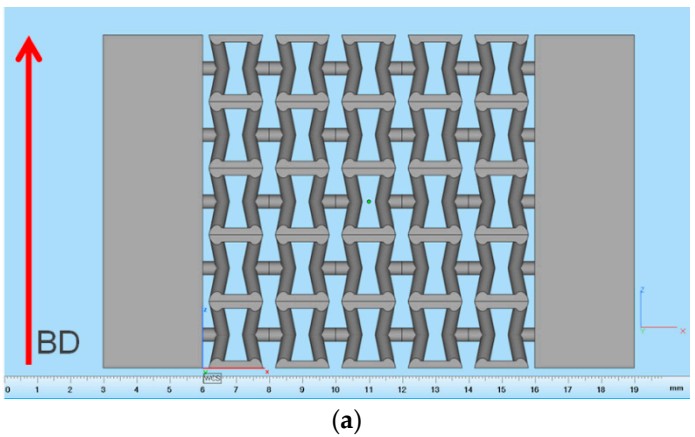
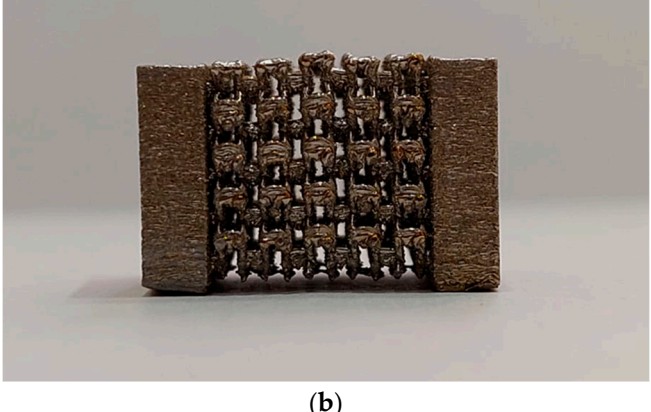

(**a**)
(**b**)

**Figure 5.** (**a**) Orientation of lattice structure based on a single Type 5 cell with bases during fabrication by the SLM technique, with the building direction (BD) indicated; (**b**) Fabricated lattice structure based on a single Type 5 cell.

Figure 6 shows SEM images of lattice structures based on a single Type 5 unit cell. You can clearly see many satellites of powder stuck to the structure struts. This may further affect porosity of the lattice structure. Linear dimensions of samples (lattice structure without bases): 10.3 mm × 10.5 mm × 10 mm. Average diameter of the lattice structure struts was 418.8 µm, which is almost equal to the strut thickness determined earlier by simulation. Porosity of the structure was 69.8%, which was 11% lower than the value assumed during simulation. This may be primarily due to the powder satellites stuck to the structure struts, as well as to some thickening of the struts in the lateral projection of the structure.

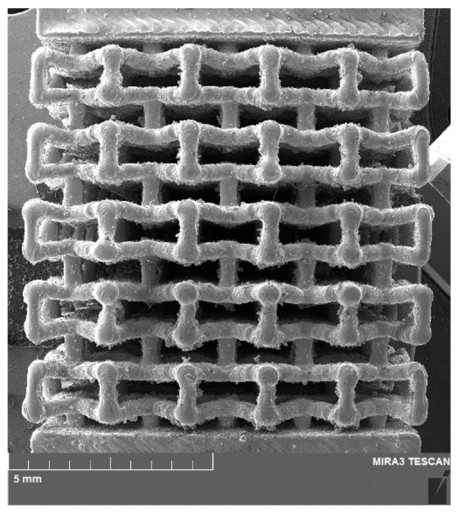
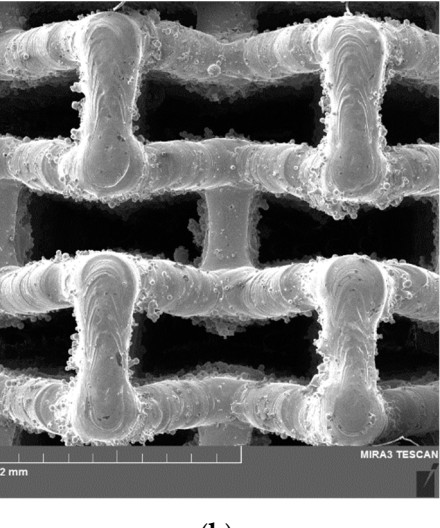

(**a**)
(**b**)

**Figure 6.** SEM images of the fabricated lattice structure based on a single Type 5 cell. (**a**) General view; (**b**) Highly magnified view of two cells.

The fabricated structures were further studied for phase composition and mechanical and functional properties.

### 3.2. Phase Composition and Transformation Temperatures

Figure 7 shows XRD patterns of the initial powder and the fabricated lattice structure sample. According to the XRD patterns, only the high-temperature B2 phase (austenite) with a cubic structure is observed in the initial powder. The fabricated lattice structure sample also

mainly contains the high-temperature B2 phase (austenite) with a cubic structure. At the same time, there was a small amount of the low-temperature martensite phase B19′, which had not been previously contained in the initial powder. Secondary phases were not detected.

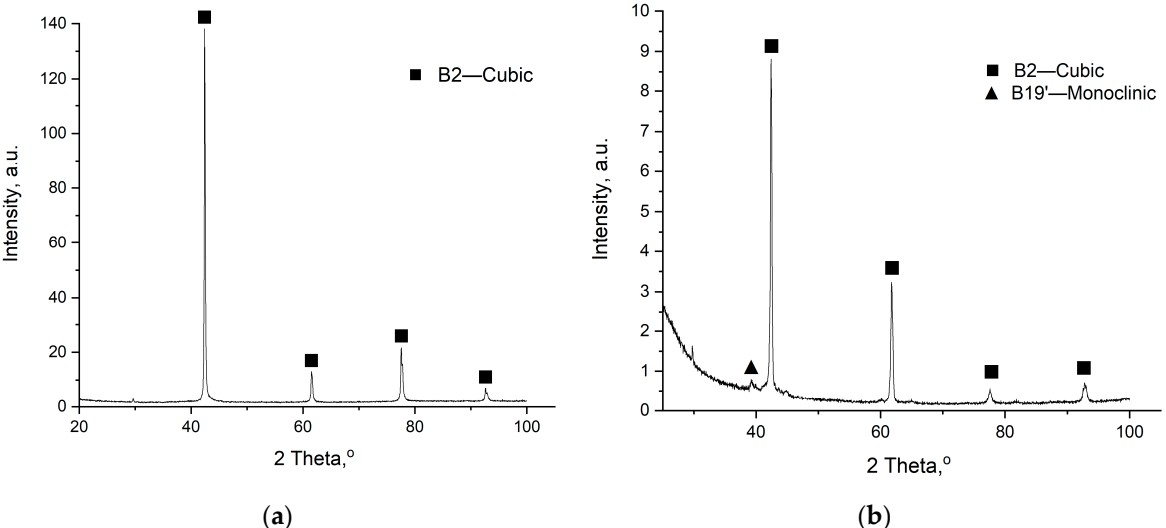

**Figure 7.** XRD patterns. (**a**) Initial powder of TiNi alloy; (**b**) A lattice structure sample fabricated by the SLM technique.

The martensite phase content indicates a possible rise in transformation temperatures due to nickel evaporation during printing, which could lead to incomplete transformation at room temperature. The phenomenon of nickel evaporation during SLM has been described in previous papers [41–45]. This assumption is validated through the DSC curves (Figure 8) plotted for the initial powder and fabricated lattice structure during heating.

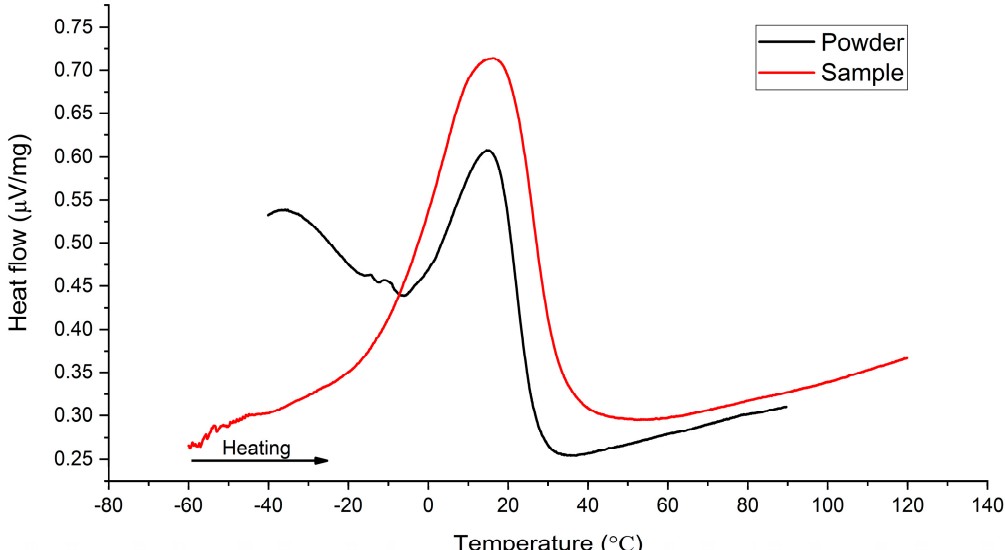

**Figure 8.** DSC curves for initial powder and lattice structure sample fabricated by the SLM technique.

The values of transformation temperatures for the initial powder and fabricated lattice structure sample are given in Table 1.

**Table 1.** Transformation temperatures.

|  | $A_s$, °C | $A_p$, °C | $A_f$, °C |
|---|---|---|---|
| Powder | −3 | 15 | 28 |
| Sample | −10 | 16 | 34 |

Table 1 shows that the peak and end temperatures of the transition to the high-temperature phase ($A_p$ and $A_f$) of the sample are clearly higher than those of the initial powder. This, in particular, can explain the martensite phase in the phase diagram of the sample: the sample is not completely in the high-temperature austenite phase at room temperature. As noted earlier, such variations are most likely due to partial evaporation of nickel in the sample printing process. At the same time, we can note some drop in the sample temperature $A_s$ as compared to the initial powder temperature.

### 3.3. Mechanical Properties

Figure 9 shows the stress–strain curves plotted in the course of mechanical compression tests of three lattice structure samples. The test results, i.e., yield strength, ultimate compression strain, Young's modulus and failure strain, an average value of each parameter, and standard deviation are given in Table 2.

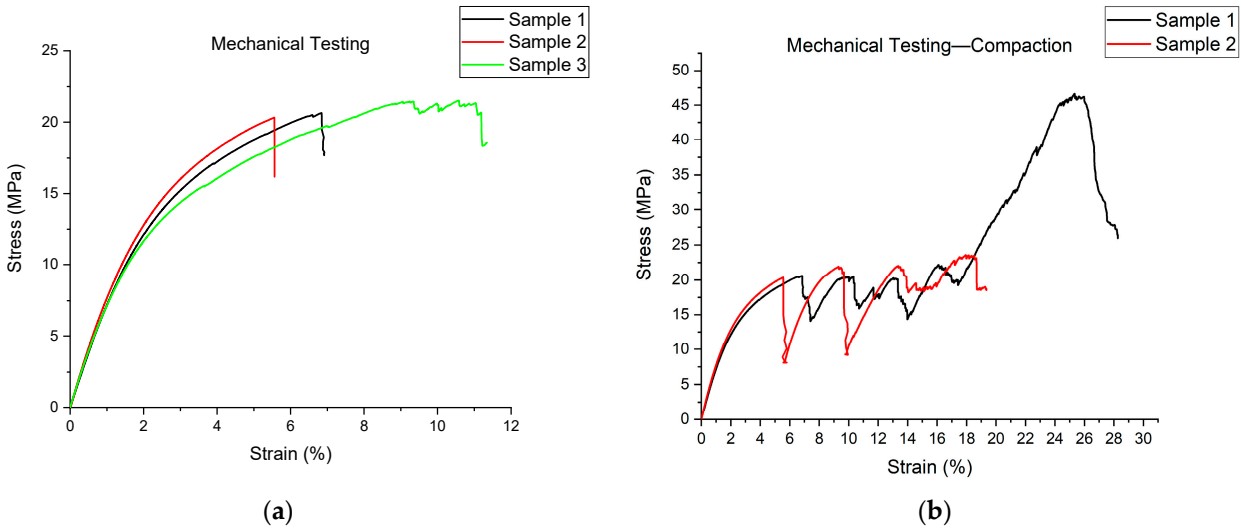

(**a**)　　　　　　　　　　　　　　　　　　　　　　　　　(**b**)

**Figure 9.** Mechanical compression test results: (**a**) Stress–strain curves for three samples plotted before initial failure; (**b**) Stress–strain curves for two samples plotted during compaction.

**Table 2.** Mechanical properties of lattice structure samples.

|  | Elastic Modulus, GPa | Yield Strength, MPa | Ultimate Comp. Stress, MPa | Failure Strain, % |
|---|---|---|---|---|
| Sample 1 | 0.796 | 9.55 | 20.97 | 6.84 |
| Sample 2 | 0.838 | 10.14 | 21.57 | 5.55 |
| Sample 3 | 0.820 | 9.4 | 22.48 | 9.33 |
| Average | 0.818 | 9.70 | 21.67 | 7.240 |
| Standard deviation | 0.021 | 0.391 | 0.760 | 1.921 |

The resulting average value of the Young's modulus of the structure given in Table 2 is a bit higher than the initial value obtained during simulation (0.61 GPa)—approximately by 0.2 GPa. First of all, this can be accounted for a lower porosity of the fabricated structures: 69.8% porosity vs. 80% porosity assumed during simulation. At the same time, the obtained Young's modulus values fully confirm the possibility to fabricate a structure with

programmable low values of the Young's modulus. The ultimate compression stress was 21.67 MPa on the average, while the first sample had a minimum value of 20.97 MPa. In this respect, the samples were further cycled to determine the potential for superelasticity effect under a maximum load of 20 MPa.

An important feature of the lattice structures with this single cell topology is their compression to failure behavior, namely, compaction. You can clearly see that on the stress–strain curve (Figure 9b). It is clearly visible that, after an initial failure of one or several cells accompanied by a voltage drop, the voltage continues rising until the next failure. This compresses and compacts the sample, causing the stress to increase even more, which is clearly visible in the stress–strain curve for sample 1 (Figure 9b). Such compaction of the sample results in its complete failure. Figure 10 shows SEM images of sample 2 with partially fractured cells. It should be noted that despite compaction, the compression failure of the sample actually occurs in the same way as for the classical material, i.e., at an angle of 45°. This is clear from Figure 10a, which shows that a line can be drawn through the fractured cells from the upper left corner of the lattice structure to the lower right corner.

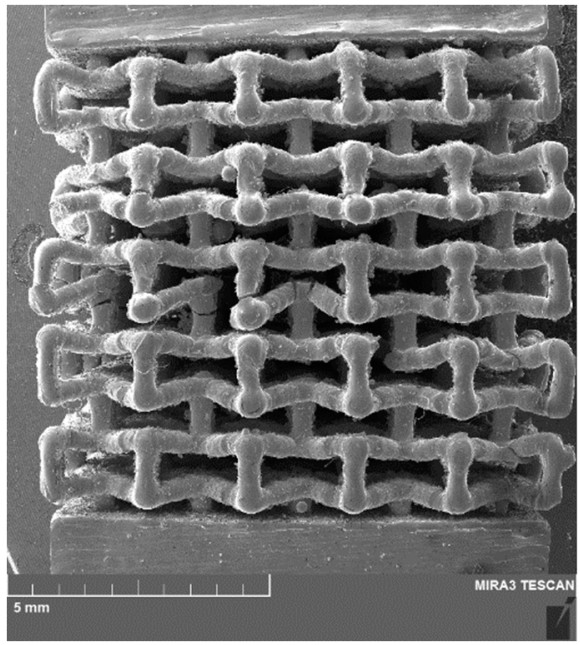

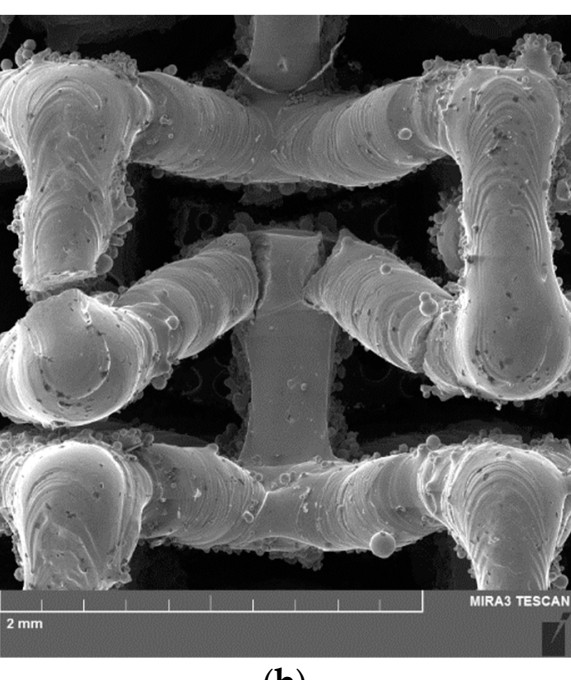

(**a**)　　　　　　　　　　　　　　　　　　　　　　　(**b**)

**Figure 10.** SEM images of sample 2 after compression to failure tests: (**a**) General view of lattice structure with visible fractures; (**b**) Partially fractured central cell.

### 3.4. Poisson's Ratio

Figure 11 shows the curves of a variation in length and width of the lattice structure (δl and δb) as a function of compression stress. It is well seen that as the longitudinal size of the sample decreases, the transverse size of the sample decreases as well. This indicates that this lattice structure has a negative Poisson's ratio. Figure 11 shows that compression was continued up to a stress of 14 MPa. Values δl and δb, being approximately equal to the average value of yield strength, were taken to calculate a final value of Poisson's ratio. According to formula (2), the lattice structure Poisson's ratio was equal to $\nu = -0.00323$, which confirms the possibility to fabricate TiNi alloy lattice structures with negative Poisson's ratio by the SLM technique. However, there is a very strong divergence in the Poisson's ratio values obtained during simulation and at final determination after the compression tests.

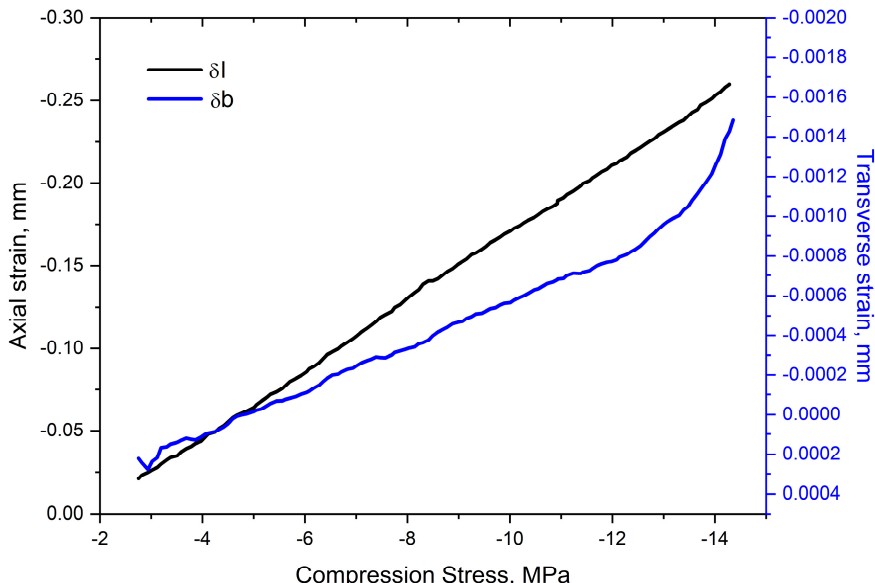

**Figure 11.** A variation in length and width of the lattice structure (δl and δb) during compression.

*3.5. Superelasticity*

The samples were cycled under various stresses at room temperature in order to determine their potential for a superelasticity effect. Figure 12a shows the results of sample cycling during seven cycles with stress continuously rising from 9 MPa to 20 MPa. Figure 12b shows five cycles for the same sample cycled under the stress of 20 MPa. A further increase in stress was found to be impractical since it would result in failure of the sample. The curves were plotted taking into account the accumulated residual strain, so each subsequent curve starts at the end point of the previous one. The last cycle under the stress of 20 MPa shown in Figure 12a is the first cycle in Figure 12b. The preload for each cycle was 0.9 MPa. The curves are plotted without regard to preload.

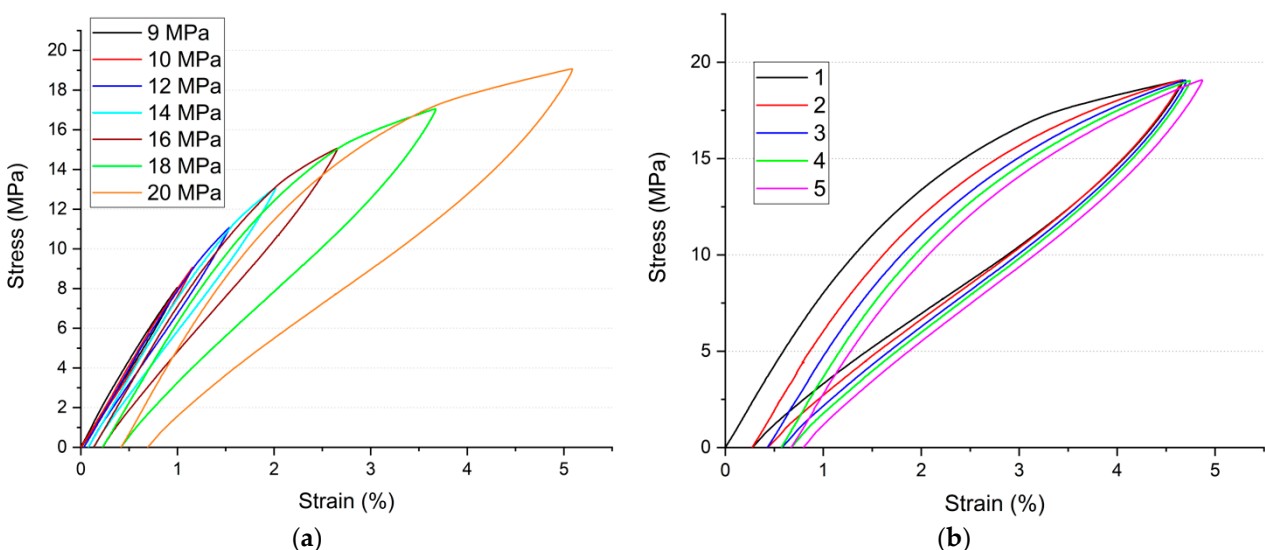

**Figure 12.** Sample cycling by compression loading: (**a**) Cycling with stress rising from 9 to 20 MPa; (**b**) five cycles under 20 MPa.

According to Figure 12, the lattice structure sample can exhibit superelastic properties at room temperature. At the same time, an extremely weak accumulation of irreversible strain can be noted. The maximum sample strain equal to 4.671% was achieved in the first cycle under the stress of 20 MPa. The maximum irreversible strain equal to 0.276% was

also achieved in the first cycle under the stress of 20 MPa. The final values of maximum ($\varepsilon_{max}$) and recoverable (reversible)/irreversible ($\varepsilon_{rev}/\varepsilon_{irr}$) strain are given in Table 3.

**Table 3.** Resultant strain achieved during cycling.

| Cycle Number | Stress, MPa | $\varepsilon_{max}$, % | $\varepsilon_{rev}$, % | $\varepsilon_{irr}$, % |
|---|---|---|---|---|
| 1 | 9 | 0.999 | 0.993 | 0.006 |
| 2 | 10 | 1.152 | 1.125 | 0.027 |
| 3 | 12 | 1.503 | 1.458 | 0.045 |
| 4 | 14 | 1.935 | 1.878 | 0.057 |
| 5 | 16 | 2.520 | 2.430 | 0.090 |
| 6 | 18 | 3.447 | 3.255 | 0.192 |
| 7 | 20 | 4.671 | 4.395 | 0.276 |
| 8 | 20 | 4.380 | 4.224 | 0.156 |
| 9 | 20 | 4.269 | 4.125 | 0.144 |
| 10 | 20 | 4.167 | 4.071 | 0.096 |
| 11 | 20 | 4.197 | 4.071 | 0.126 |
| Total amount | | | | 1.215 |

## 4. Discussion

The fabricated lattice structures, in terms of their properties, matched the initial simulation results in general. Deviations in porosity of the structures (69.8% porosity vs. 80% assumed in the simulation) are a specific feature of the fabrication technique and the SLM system itself. This also can explain some excess in the resultant values of elastic modulus specified in Table 2. It can be noted that the obtained values fall within the elastic modulus interval for a human trabecular bone. However, porosity of the fabricated lattice structure should be reduced to achieve the proper elastic modulus of a cortical bone. In addition, the obtained values of the porosity and modulus of elasticity of lattice structure are lower than the values obtained in other works. For example, in [32], the smallest Young's modulus was 6.4 GPa with the porosity of the structure 59.09%. In [24], it was 9 GPa with the porosity 58%, and in [30], the lowest value of the modulus of elasticity was 16.5 GPa with the porosity 65%. It should be noted that the compression deformation is rather low, averaging less than 22 MPa. This value is lower than that of a human cortical bone for which it is estimated around 153–205 MPa [21]. The resultant negative values of the Poisson's ratio are numerically very different from the values obtained during simulation. In this case, the obtained results are probably influenced both by a higher porosity and by errors that may arise when determining strains of a sample.

As previously noted, a variation in the alloy transformation temperatures was caused by evaporation of nickel from the TiNi alloy powder during SLM process. Intensive evaporation of nickel is due to the difference between the boiling points of nickel and titanium: nickel has a boiling point of 2913 °C, while the titanium boiling point equals 3287 °C. It is known that a variation in nickel content in the TiNi alloy greatly affects the transformation temperatures of the alloy. A 0.1 at. % increase in nickel content leads to a 10 K decrease in the transformation temperatures. On the contrary, a decrease in nickel content leads to an increase in the transformation temperatures [46–48]. Accordingly, a decreased nickel content in the alloy after lattice structure fabrication by the SLM technique led to an increase in $A_p$ and $A_f$ temperatures.

The studies of the samples' superelasticity showed a rather high durability of the sample under high loads. The total accumulated irreversible strain amounted to 1.2% after 11 cycles (see Table 3) provided that the last five cycles were conducted under a load being close to the ultimate compression strain. The test temperature was lower than temperature Af of the fabricated lattice structures (see Table 1), which shall also negatively affect the possibility of reversible strain, and lead to incomplete recovery and an increase in irreversible strain.

The curve of strain values as a function of the cycle (Figure 13) was plotted on the basis of the data in Table 3. According to the figure, both reversible and irreversible strains rise proportionally to stress increase. As noted earlier, the maximum value of all strains was obtained during cycle 7. After that, despite the maintained stress of 20 MPa, there was a slight drop in the maximum, reversible, and irreversible strain. This may be due to stabilization of the sample superelastic response. The stabilization effect of shape memory and superelasticity response of the SLM structures was previously noted in several works [24,30,33].

Some limitations of this work should be noted. Firstly, the computer simulation did not consider the behavior of the lattice structure in the plastic region due to the need to determine the characteristics only in the elastic zone. Secondly, only one unit cell topology was considered in this work. Thirdly, only a part of the transformation temperatures of alloy is determined in the work, due to the technological features of the equipment. On the issue of accuracy, one can note the general repeatability of the results obtained for the samples. Moreover, the samples should be further tested by the cycling method but at different temperature conditions, namely at a temperature close to human body temperature: 36.6–37 °C. At the same time, it should be taken into account that the results of the experiments, when repeated, may vary depending on the initial powder (its quality, chemical composition), the SLM system used, and the parameters of the manufacturing process.

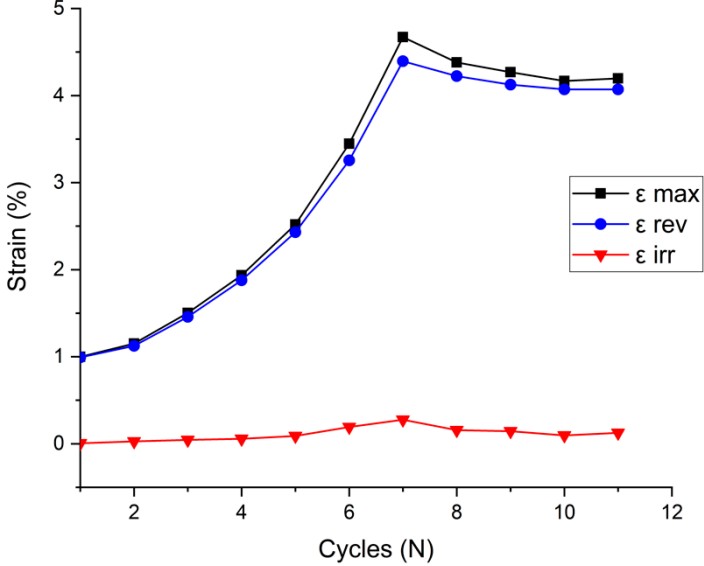

**Figure 13.** Maximum, reversible, and irreversible strain as a function of cycle.

## 5. Conclusions

Several main conclusions can be drawn from the studies and obtained experimental data presented in this paper. Firstly, the obtained experimental data mainly match the data obtained during simulation. So, it will be possible to use this "simulation experiment" link in the in further works. Secondly, the technical and practical feasibility of fabricating highly porous lattice structures with low Young's modulus values and negative Poisson's ratio by the SLM technique was confirmed. Finally, the fabricated lattice structures are capable of realizing the superelasticity effect at room temperature under a load being close to the limit load, with a low level of accumulated irreversible strain.

The data obtained confirm the operability of the presented complex method for obtaining lattice structures with programmable properties. The possibility to fabricate lattice structures with negative Poisson's ratio and a Young's modulus value closer to the elastic modulus of a human cortical bone will be discussed in subsequent papers. Moreover, the subsequent papers will cover the method of modifying the lattice structure geometry to obtain greater values of negative Poisson's ratio. Future works will be a logical continua-

tion of the development of the concept of obtaining lattice structures from TiNi alloy with programmable properties using computer simulation and the SLM method.

**Author Contributions:** Conceptualization, E.F. and A.O.; methodology, E.F. and A.O.; software, A.O. and A.R.; formal analysis, E.F.; investigation, E.F., N.G. and S.K.; writing—original draft preparation, E.F.; writing—review and editing, E.B.; project administration, A.P.; funding acquisition, E.F. and A.P. All authors have read and agreed to the published version of the manuscript.

**Funding:** The reported study was funded by RFBR according to the research project № 20-38-90031.

**Institutional Review Board Statement:** Not applicable.

**Informed Consent Statement:** Not applicable.

**Data Availability Statement:** The data presented in this study are available on request from the corresponding author.

**Conflicts of Interest:** The authors declare no conflict of interest.

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
