# Peer review of "TiNi Alloy Lattice Structures with Negative Poisson’s Ratio: Computer Simulation and Experimental Results"

_metals, doi:10.3390/met12091476_

Round 1

Reviewer 1 Report

This manuscript studied the TiNi alloy lattice structures with negative Poisson’s ratio using both simulation and experimental approaches. The results and analysis provided by the authors are reasonable and rigorous. Some revisions need to be considered as follows:

1.      There are some grammar mistakes and the language need to improve.

2.      Page 2, what is the CNPC?P lease give the full name instead of the abbreviation when it shows initially in the manuscript.

3.      The legend and the coordinate axis in Fig. 2 are too small and blur.  

4.       There are several different scales and level structures based on the same Type 5 single cell, such as 2*2*2 and 5*5*5 Lattice structures as shown in Fig 2. Readers may confuse in the following results section about the single cell and 5*5*5 structure. The proper name should be assigned for the structure as well.

5.      In the section on Materials and Methods, some detailed simulation parameters are referred to in the previous paper [24], this is not friendly to readers. Please give the basic simulation conditions such as load type, boundary conditions, mesh strategy, and size.

6.      It seems that only the linear part of the stress-strain relationship was considered in the simulation process as given by the properties of the material from lines 99 to 101, which was the elastic relation. But the experimental results of the negative Poisson’s ratio structure included both the elastic and the plastic relation. Why the author did not consider the plastic behaviors and the work-hardening behaviors?

7.       Line 144 on page 4, what means the thickness of the lattice structure struts? Thickness seems not a proper word to describe the structs. Is it the diameters of the struts?

8.       Figure 6 is not the phase diagram, it is XRD results, which give the phase information. The term phase diagram is the composition vs temperature diagram of two or multiple elements.

9.       Page 6, why the existence of martensite means nickel evaporation? What is the relationship between the transformation temperature and nickel evaporation? The melting temperatures of Ti and Ni are very close, it only has a 200-degree difference. How can you verify only nickel was evaporated during the SLM process? Is some of the Ti also evaporated as well?

10.    In the conclusion part, the second and the third conclusions can be combined into one sentence.

Author Response

Thanks for your review. It allowed us to improve our work.

Reviewer 2 Report

The manuscript presents the simulation and fabrication by the SLM technique and study of NiTi lattice structures with negative Poisson’s ratio.

The study lacks novelty as the fabrication of porous NiTi alloys with negative Poisson’s ratio with SLM has been done before and is a known idea within the community. The authors must clarify what is novel in this manuscript.

Many key experimental details are missing in M&M:

1. the simulation procedure, although it has been published before, please give a distillation of it. If this Type 5 is from ref 24, can we assume the simulation results are not new?

2. Compaction test

3. SEM

4. XRD

5. Cyclic loading

The claimed negative Poisson’s ratio and low Young's modulus were the outcome of the tests at room temperature. Do the structures still process these at the application temperature of 37 celsius?

Why are results Young's modulus (E) – 0.61 GPa, shear modulus (G) – 3.68 GPa, Poisson's ratio (ν) – -0.13 different from the initial simulation results?

The references are outdated: for “ In dentistry, the alloy is used as braces,” please cite the latest research article on TiNi for dentistry published in the same journal e.g. 10.3390/met12030406

Further, there lacks explanation of underlying mechanism of the observed negative Poisson’s ratio and low Young's modulus at room temperature.

Fig. 10, strain should not have unit.

Author Response

Thanks for your review. We hope that all fixes will meet your requirements.

Reviewer 3 Report

1.      The current title should have both the uppercase and lowercase letters changed to conform to MDPI format.

2.      It is needed to provide all of the Author’s email after affiliation (with initials author nafe if more than 1) following MDPI format, except for corresponding.

3.      Quantitative results need to be added in the abstract section.

4.      Please end your abstract with a "take-home" message.

5.      Rearrange keywords alphabetically.

6.      Please, use the lowercase font for each of the keywords according to MDPI format.

7.      Abbreviation as a keyword is not recommended and encouraged to be changed become a stand for its abbreviation, see SMA in line 18.

8.      What makes the author's novelty in the present work? My analysis suggests that other similar previous studies properly explain the points you have brought up in the current paper since the negative Poisson's ratio has been widely discussed in the literature. Please be sure to emphasize anything truly novel in this work in the introductory section.

9.      In order to demonstrate the research gaps that the current study aims to address, previous studies linked to it need to be explained in the introduction part, including their work, their novelty, and their limitations.

10.   In the introduction section, the authors need to explain several approaches of analytical, computational, and experimental study, their advantages and their limitation. It is important to point out that need to be included by authors in the introduction and/or discussion section. Also, to support this explanation, the suggested reverence published by Metals, MDPI should be adopted as follows: In Silico Contact Pressure of Metal-on-Metal Total Hip Implant with Different Materials Subjected to Gait Loading. Metals (Basel). 2022;12(8):1241.

11.   Related to previous comments, it is needed to explain why the present study performs only computational and experimental study.

Author Response

Thanks for your review. We hope our fixes meet your requirements.

Round 2

Reviewer 1 Report

The manuscript can be published as this revised version.

Author Response

Thank you very much for your review.

Reviewer 2 Report

After extensive revision addressing all the points raised, it is ready for publication.

Author Response

Thank you very much for your review.

Reviewer 3 Report

Good job to the authors, I have several comments that needs to be addressed/

1.      To enhance the understandability of the section on materials and methods easier for them to understand rather than just depending on the main text as it exists at the moment, the authors could add additional illustrations in the form of figures that explain the workflow of the present study.

2.      Other information about the tool, such as the manufacturer, country, and specifications, should be provided.

3.      Error and tolerance of experimental tools used in this work are important information that needs to be explained in the manuscript. It is would use as a valuable discussion due to different results in further study by another researcher.

4.      A comparison of the results with similar past investigations is required.

5.      It is needed to mention regarding biocompabiliy aspect of titanium alloy materials that make it suitable for implant materials. It is a vital topic that authors must provide in the introduction and/or discussion section. Additionally, the suggested reverence should be taken to substantiate this explanation as follows: Tresca Stress Evaluation of Metal-on-UHMWPE Total Hip Arthroplasty during Peak Loading from Normal Walking Activity. Mater. Today Proc. 2022, 63, S143–6. https://doi.org/10.1016/j.matpr.2022.02.055

6.      What is the limitation of the present work? Please include it before the conclusion section.

7.      Elaborate the conclusion as a form of paragraph, not point by point as present form.

8.      Mention further research in the conclusion section.

9.      Five years back literature should be enriched into the reference, and MDPI-published literature is highly recommended.

10.   The authors occasionally created paragraphs in the entire document that were just one or two phrases long, which made the explanation difficult to understand. To make their explanation into a longer, more thorough paragraph, the authors should expand it. It is advised to use at least three sentences in a paragraph, with one serving as the primary sentence and the others as supporting phrases. For example, the second paragraph of introduction section.

11.   Due to grammatical problems and linguistic style, the authors should proofread the work. It would be used MDPI English editing service for this concern.

12.   Ensure that the authors followed the MDPI format exactly, edit the current form, and double-check all of the previously noted problems.

Author Response

Thank you for your work. Please see the attachment.
